# Indoleamine 2,3-Dioxygenase as a Therapeutic Target for Alzheimer’s Disease and Geriatric Depression

**DOI:** 10.3390/brainsci13060852

**Published:** 2023-05-24

**Authors:** Karl Savonije, Autumn Meek, Donald F. Weaver

**Affiliations:** 1Krembil Research Institute, University Health Network, Toronto, ON M5T 0S8, Canada; 2Departments of Chemistry and Medicine, University of Toronto, Toronto, ON M5S 3M2, Canada

**Keywords:** indoleamine 2,3-dioxygenase, tryptophan, neuroinflammation, depression, dementia, Alzheimer’s disease

## Abstract

Neuroimmune-triggered neuroinflammation of the central nervous system is emerging as an important aetiopathogenic factor for multiple neurological disorders, including depression, dementia, Alzheimer’s disease, multiple sclerosis and others. Tryptophan metabolism via the kynurenic pathway, which is initiated by the indoleamine-2,3-dioxygenase (IDO-1) enzyme, is a key regulator of the neuroimmune system and its associated neuroinflammatory effects. As discussed in this review, targeting the production of immunopathic and potentially neurotoxic kynurenine metabolites by inhibitory downregulation of IDO-1 may prove a viable target against inflammation-induced neurological conditions, particularly depression and dementia.

## 1. Introduction

Tryptophan (Trp) is an essential amino acid that is centrally implicated in the molecular pathogenesis of many neurological and psychiatric conditions, including dementia and major depressive disorder (MDD) [1]. Trp is metabolized into serotonin (5-hydroxytryptamine, 5-HT) via the initial conversion to 5-hydroxytryptophan, catalyzed by tryptophan hydroxylase, and then to 5-HT, which is catalyzed by aromatic amino acid decarboxylase. 5-HT is a well-studied neurotransmitter that is targeted by multiple potent psychotropic agents, ranging from therapeutic antidepressant selective serotonin reuptake inhibitors (SSRIs) to recreational serotonin-releasing agents (SRAs), such as amphetamines and tryptamines. Despite many of these having been exploited historically as drug targets for psychiatric disorders, 5-HT is, in fact, a minor metabolite of Trp, with the majority (>90%) of Trp being metabolized via the kynurenine pathway [2].

The kynurenine pathway (KP) generates a variety of metabolites, collectively referred to as kynurenines. These metabolites are important for immune signaling, neurological modulation and the generation of other biologically active compounds, including nicotinamide adenine dinucleotide (NAD+) [3]. The first metabolic step in the KP is the rate-limiting conversion of Trp to kynurenine (KYN), normally initiated in the liver as a hepatic process that is primarily catalyzed by the tryptophan 2,3-dioxygenase (TDO) enzyme [4]. However, the KP can be initiated extra-hepatically primarily by indoleamine 2,3-dioxygenase (IDO-1, IDO-2) [4]. Under pathological immune stimulating conditions, IDO-1 is an inflammatory immune signaling enzyme whose expression is upregulated by pro-inflammatory cytokines, particularly interferon-gamma (IFN-γ) and tumor necrosis factor-alpha (TNF-α), and can be produced in most biological tissues [5,6,7]. KYN and its downstream metabolite 3-hydroxykynurenine (3-HK) are plasma transported and readily cross the blood–brain barrier (BBB), such that approximately 60% of all brain KYN is, in fact, initially generated in the periphery [8]. The KYN metabolism in the central nervous system (CNS) and periphery is similar. Although a portion of CNS KYN may be metabolized to the neuroprotective kynurenic acid (KYNA) in microglia and astrocytes by kynurenine transferases (KATs), the bulk is preferentially hydroxylated to 3-HK and progresses onward in a catabolic cascade to the eventual production of quinolinic acid (QUIN) and picolinic acid. QUIN has been identified as a neurotoxic agent, and an elevated QUIN level is associated with various neurological pathologies, including depression (e.g., MDD), dementia, Alzheimer’s disease (AD) and multiple sclerosis (MS) [9]. Finally, QUIN is converted to NAD+ and enters the nicotinamide cycle.

Thus, three heme-containing enzymes, IDO-1, IDO-2 and TDO, catalyze the initial rate-limiting step of Trp metabolism, leading to the cascade of kynurenine metabolites. Of these three enzymes, IDO-1 is the one with the most data supporting its upregulation during pathological pro-inflammatory conditions. Preclinical and clinical studies support that IDO-1 overexpression is associated with a poor prognosis across a spectrum of pathologies and is best demonstrated in cancer [10]. IDO-1 is also widely expressed throughout the brain, particularly in the hippocampal and limbic structures centrally involved in AD [11]. IDO-2, on the other hand, is expressed in the liver, kidney, and male and female reproductive systems. Moreover, as a catalyst, IDO-2 is less efficient and effective than IDO-1 [12]. TDO is also heavily expressed in the liver but has been identified in the brain and even co-localized with quinolinic acid, neurofibrillary tau-tangles and amyloid deposits in the hippocampi in both transgenic AD mouse models and human subjects [13]. Because of its well-described upregulation during pathology, brain localization and involvement in AD, IDO-1 has been identified as a possible druggable target for neurodegenerative disorders. A full appreciation of TDO’s participation in neuropathology is still unfolding.

Since inflammatory conditions upregulate the activity of IDO-1, Trp is increasingly diverted to the KP over the 5-HT pathway in areas of inflammation [14]. Early hypotheses of an inflammatory model of MDD postulated that in chronic or systemic cases of inflammation, this diversion of Trp metabolism would cause a relative 5-HT deficiency in the CNS, explaining the onset of depression symptoms. However, more recent data suggest that CNS Trp and 5-HT concentrations in the brain actually increase in response to inflammatory challenges, and thus, there are other mechanisms at play to regulate 5-HT [15]. However, ongoing data do support a pathogenic role for IDO-1 in MDD. Severe depression is associated with increased microglial quinolinic acid in subregions of the anterior cingulate gyrus—possible evidence of immune-modulated glutamatergic neurotransmission dysregulation [16,17]. The IDO-1 blockade in lipopolysaccharide (LPS)-induced rodent neuroinflammation models has shown a reduction in stress and depressive behaviors [15,18,19].

This raises the possibility of IDO-1 being a broadly applicable therapeutic target for neuroinflammation-associated brain disorders, including AD and related dementias, as well as MDD. Since IDO-1 is also a target of interest in neuro-immuno-oncology, in principle, lessons learned from the oncology experience and IDO-1 drugs developed for cancer indications could be leveraged as an advanced starting point in the evaluation of the therapeutic utility of IDO-1 inhibition for CNS disease indications, such as AD and elderly onset depression.

## 2. Tryptophan Metabolism

Trp is an essential amino acid that higher trophic lifeforms are incapable of producing in vivo and must be acquired from dietary sources. Trp is required for protein formation, and is catabolized into a variety of bioactive metabolites, primarily via two distinct metabolic pathways: the serotonin pathway leading to melatonin and the kynurenine pathway leading to the nicotinamides (See Figure 1). Each pathway produces biologically important regulatory chemical intermediates that are involved in regulating neural function, immune responses and metabolism.

### 2.1. Serotonergic Pathway

The serotonin pathway is the lesser of the two pathways responsible for generating the 5-HT neurotransmitter: Trp is hydroxylated to 5-HTP by tryptophan hydroxylase (TPH) and then decarboxylated by aromatic acid decarboxylase (DDC) to 5-HT. Since the conversion of 5-HTP to 5-HT is very rapid, TPH-mediated hydroxylation is the rate-limiting step in 5-HT production [20]. 5-HT synthesis is largely confined to the CNS, although the gut microbiome may also be an important source of 5-HT [21]. 5-HT is eventually converted into the hormone melatonin, which is involved in circadian rhythm regulation and is widely used/abused as a readily available sleep-inducing agent.

### 2.2. Kynurenic Pathway

This pathway produces a variety of biologically important kynurenic compounds (Figure 1). The first and rate-limiting step is the conversion of Trp to *N*-formylkynurenine by either IDO-1 or TDO, followed by the rapid *N*-formylkynurenine formamidase-catalyzed conversion to KYN [22]. KYN metabolism in the CNS follows two pathways: astrocytes convert KYN to KYNA using the kynurenine aminotransferases (KATs), while microglia convert KYN to 3-HK using kynurenine monooxygenase (KMO). 3-HK is converted into 3-hydroxyanthranilic acid (3-HAA) and then into QUIN, which is eventually catabolized into nicotinamide, an important metabolic coenzyme. KYN metabolites are functionally complex from a neurochemical perspective; some are putatively neurotoxic (3-hydroxykynurenine (3-HK) and quinolinic acid (QA)) while others are reputedly neuroprotective (kynurenic acid (KA). 3-HAA is another multifunctional KYN metabolite that exhibits potentially useful neuropharmacologic properties, such as inhibiting aberrant protein misfolding and oligomerization (e.g., β-amyloid) and downregulating pro-inflammatory cytokines.

## 3. Function and Structure of IDO

IDO-1 also plays a definite role in immune signaling and, as described in mouse models, acts as a pro-inflammatory B lymphocyte mediator [12,23]. IDO-1 is synthesized as an immune signaling response in situ by multiple cell types [5,7]. The presence of pro-inflammatory cytokines, such as TFN-α, IFN-γ and IL-6 (interleukin-6), upregulates the expression of IDO-1 and decreases the local Trp:KYN ratio. This, in turn, activates T-regulator immune cells, suppressing T-effector cells and dampening the immune response, which may lead to immune escape, as evidenced by the unusually high expression of IDO-1 in immune-avoidant tumors [24,25].

The central role of IDO-1 in regulating the immune responses is orchestrated by both enzymatic and non-enzymatic routes. From an enzymatic perspective, the IDO-1-triggered kynurenic pathway contributes to immunoregulation via five mechanisms [26]: (a) by the kynurenic-mediated inhibition of IL-2 signaling, impairing memory CD4 T-cell survival; (b) by promoting the differentiation of CD4 T-cells into Treg cells; (c) by the direct effect of kynurenines on the aryl hydrocarbon receptor (AhR), stimulating dendritic cell differentiation; (d) by modulating Trp via the induction of the stress response kinase, general control nondepressible 2 (GCN2), and the suppression of the mammalian target of the rapamycin 1 (mTOR1) pathway, which inhibits Teff-cell function and maturation; and (e) by the inactivation of the eukaryotic translation initiation factor 2A (eIF2A), which blocks the conversion of Treg cells into pro-inflammatory T-helper-type 17 (Th17) cells. The non-enzymatic immunoregulatory function of IDO-1 depends on the presence of two immunoreceptor tyrosine-based inhibitory motifs, ITIM1 and ITIM2, located in the enzyme’s non-catalytic domain [27]. When tyrosine is phosphorylated, IDO-1′s ITIM sequence becomes a docking site for various molecular partners containing Src homology 2 (SH2) domains, which enables Src homology 2 domain phosphatase-1 (SHP1) and SHP2 to interact with the interleukin-1 receptor-associated kinase (IRAK), thereby activating noncanonical, anti-inflammatory NF-κB rather than canonical, pro-inflammatory NF-κB; in turn, noncanonical NF-κB, associated with activated inhibitory-κB kinase α (IKKα), translocates into the nucleus, inducing the *IDO1* gene and subsequently establishing a positive feedback loop and granting a long-term immunoregulatory phenotype.

IDO-1 is part of a family of enzymes, including IDO-1, IDO-2 and TDO, which act to convert Trp to KYN. TDO is a tetrameric enzyme primarily produced in the liver and kidneys and is responsible for the majority of the hepatic Trp conversion as part of Trp metabolic homeostasis [28]. IDO-1 and IDO-2 are monomeric enzymes that share strong structural similarities with each other but not with TDO [23]. Despite the structural similarities, IDO-2 shows decreased efficacy in the catabolizing of Trp compared to IDO-1 and is expressed primarily in the liver and kidneys [23], unlike IDO-1, which is expressed in a wide variety of different tissues, including the brain [5,7]. (See Figure 2).

Human IDO-1 is a heme-centered protein comprising 403 amino acid residues. The N-terminal domain (NTD) contains the first 154 residues and is constructed of six α-helices, two small β-sheets and three 3_10_ helices [29]. The larger C-terminal domain (CTD), containing residues 155–403, is located below the NTD and comprises thirteen α-helices and two 3_10_ helices [29]. In the CTD, four long helices, conventionally designated G, I, Q and S, run in parallel to the heme plane and, together with the side chains of the K, L and N helices, form the heme-binding pocket. In the presence of the primary substrate, Trp, the C-terminal end of the disordered JK-loop organizes into a β-hairpin structure [29]. The active site on the distal heme porphyrin is enclosed by the NTD and the JK and DE loops [30]. There are four binding sites: (a) Pocket “A” is a hydrophobic domain on the distal heme side and contains residues Tyr126, Cys129, Val130, Phe163, Phe164, Ser167 and Ala26 [22,31]; (b) pocket “B” sits between pocket A and the entrance to the active site and is only accessible when the JK-loop is in an open conformation [32]; (c) pocket “C” is an accessible surface at the entrance to the active site, partially obstructed by the JK-loop; and (d) pocket “D” sits within the CTD on the proximal side of the heme and is largely solvent shielded, although small molecules may be able to access it with some degree of heme displacement. Pockets D and C have both been suggested to exhibit inhibitory function.

IDO-1′s primary substrate, Trp, binds to the “A” site through a water-mediated H-bond between its indoleamine group and Ser167 [30], while the ammonium and carboxylate groups bind to the “B” site through a series of H bonds with the C-terminal end of the JK-loop. Trp is ring-opened to *N*-formylkynurenine via the insertion of heme-coordinated dioxygen over the C2–C3 bond via several proposed mechanisms [33,34].

## 4. Drugability of IDO

IDO-1 upregulation is suggested to be involved in cancer immune avoidance, making IDO inhibitors an attractive prospect for novel anti-cancer therapeutics when seeking to leverage the therapeutic benefits of immune system modulation. Accordingly, IDO-1 is a target of interest in oncology, with several drug candidates having proceeded to Phase I/II human trials in recent years [35,36]. In addition to the small molecule drug candidates, IDO-1 peptide vaccines have also been designed [37].

Preclinical oncology studies evaluating a structurally diverse range of IDO inhibitors showed promising results; for instance, in murine models of melanoma, the combination of IDO-1 blockade with checkpoint inhibitors significantly decreased xenograft growth and increased local cytotoxic T-cell proliferation [25]. Accordingly, multiple IDO-1 inhibitors were advanced to clinical trials, with epacadostat receiving extensive study. Although epacadostat exhibits over a 1000-fold selectivity for IDO-1 over IDO-2/TDO, the phase III trial of epacadostat combined with the PD-1 inhibitor pembrolizumab did not show a clinical benefit when compared with pembrolizumab monotherapy in patients with advanced malignant melanoma—this negative study elicited a general dampening of excitement regarding the therapeutic utility of IDO-1 inhibitors in cancer [38]. Other agents did exhibit some modest efficacies. For example, clinical results indicated that indoximod, when used as a single agent, exerted little antitumor efficacy, yet when used in combination with other therapies, including checkpoint inhibitors (pembrolizumab, nivolumab) and chemotherapy, indoximod demonstrated enhanced antitumor efficacy [39]. Another IDO-1 inhibitor, navoximod, combined with atezolizumab to treat locally advanced or metastatic solid tumors, produced a partial response in 9% of the dose-escalation patients, with adverse events including fatigue, rash and chromaturia [40]. Overall, in the field of IDO-1 inhibition for the treatment of malignancies, there has been a substantive mismatch between the preclinical efficacy and ultimate clinical trial outcomes, which significantly reduced the widespread development of the therapeutic class.

While it is tempting to suggest that these drugs could be co-opted for use in the treatment of neurological conditions, it is important to note that, in order to affect IDO-1 activity in the CNS, the drug molecule is required to be a brain penetrant. Brain penetrant molecules typically are of low molecular weight and lipophilic; unfortunately, the majority of existing oncology-targeted IDO-1 inhibitors exhibit low lipophilicity due to the presence of the hydrophilic moieties required for H-bonding interactions in the IDO active site. This may render the majority of existing drug candidates unsuitable for CNS-targeted therapies, although tools exist for rapidly screening drug candidates, such as the BBB Score [41] and the Central Nervous System Multiparameter Optimization calculator [42]. Drug discovery/design for CNS IDO inhibition must account for the BBB, whether through first-principles design for passive BBB penetration, novel delivery mechanisms such as nano-shells, or co-opting the active transport mechanism.

In designing and developing novel IDO inhibitors for a CNS indication, multiple approaches are possible. The derivitization of known IDO inhibitors developed for oncology indications (e.g., epacadostat, navoximod, indoximod, NLG802, LY3381916) to incorporate BBB-penetrant design features is a possibility; however, they face challenges of lost potency and efficacy when the compounds are structurally altered (See Figure 3). High-throughput screening campaigns, either in silico or in vitro, may be exploited to identify “hits”. Finally, de novo drug design may be pursued based on structural biology knowledge of the IDO protein and its competitive or allosteric receptor sites. Recently, Zheng et al. reported the structure-based design of a series of brain-penetrant novel 2-(5-imidazolyl)indole analogue IDO1 inhibitors based on preliminary studies using N1-substituted 5-indoleimidazoles [43]. By employing structure–activity relationship (SAR) studies, computational docking simulations and BBB score calculations, they designed multiple brain-penetrant compounds that interacted favorably with a proximal Ser167 residue in IDO (See Figure 4), as well as one representative analog within this series and demonstrated strong IDO1 inhibitory activity (IC50 = 0.16 μM, EC50 = 0.3 μM)—this compound has yet to progress to additional biological testing in AD models. Therefore, the design of brain-penetrant drug-like IDO inhibitors is a surmountable molecular design task.

Despite the opportunities afforded by these multiple approaches, the development of IDO-1 inhibitors for the treatment of neurological disorders such as depression or dementia must confront a myriad of efficacy and ADMET (absorption, distribution, metabolism, excretion and toxicity) design challenges. The ideal drug should have a half-life of approximately 24 h so that it may be administered once per day, which is optimal when treating mood or memory disorders. The compound should have minimal interactions with other drugs, given that the treated age group is often receiving multiple medications. Drugs used to treat chronic neurological disorders are typically taken for months or years and, thus, should exhibit few, if any, long-term toxicities. The drug must be bioavailable to receptors within the brain and, therefore, must be able to traverse the blood–brain barrier. Even if the molecule is able to enter the brain, will inhibiting IDO-1 be sufficient or should it also be able to concomitantly block TDO? Following receptor(s) binding, the compound should be truly disease-modifying and not merely provide symptomatic improvement. Finally, the available animal models for diseases, such as AD, are best described as models of protein misfolding rather than representative models of the actual disease; therefore, the ability of such animal models to accurately capture the proposed disease-producing mechanism of action of neuroimmune-triggered neuroinflammation is limited, constituting a significant impediment to compound advancement. Accordingly, there are many hurdles when attempting to achieve lead optimization and clinical candidate identification in a process that requires a proven mechanism of action and evidence of target engagement that leads to the therapeutic efficacy of IDO-1 inhibitors.

## 5. IDO and Depression

Most current first-line pharmacological therapies for depression focus on serotonin metabolism, typically by selective serotonin reuptake inhibitors (SSRIs), although some research has also considered dietary tryptophan supplements [44]. This model of depression—the monoamine depression theory—has produced a series of widely used therapeutic agents for depression, which did not have the significant side effects of the earlier generations of antidepressants and, as such, has been the dominant pharmacological approach to the treatment of depressive disorders for the latter half of the 20th century [45]. SSRIs are, however, imperfect: as many as half of all cases of depressive disorder prove resistant to treatment. Recently, a newer model for depression has been advanced that instead focuses on the kynurenine pathway and its neurotoxic metabolites [46]. It is proposed that chronic inflammatory conditions cause the overexpression of IDO-1 and correspondingly divert tryptophan metabolism away from 5-HT and towards the KP with its neurotoxic metabolites.

This inflammatory model of depression initially suggested that the increased KP activity would reduce available Trp and hence, 5-HT, inducing depressive behavior through a 5-HT deficiency [47]. Experimental evidence for this has proven inconclusive; however, plasma Trp and 5-HT titres are decreased when presented with an inflammatory challenge, and the CNS levels remain stable, even as the CNS Trp:KYN ratio decreases. Instead, it is suggested that the cause of depression is, in fact, the neurotoxic effects of key KP metabolites, namely 3-HK and QUIN. The conversion of KYN to neuroprotective KYNA is rate-limited by the KATs, the enzymes responsible for the conversion of KYN to KYNA by astrocytes; thus, any excess KYN present in the CNS will primarily follow the QUIN pathway: conversion to 3-HK and then on to QUIN, and thus increasing oxidative stress in the CNS and directly linking increased CNS KYN levels to neurotoxic conditions. This, in turn, can cause a vicious cycle whereby sustained oxidative stress causes cell lysis and triggers the release of cytokines, further stimulating the production of IDO and promoting more Trp-to-KYN conversion.

While many of these models only consider the CNS, peripheral inflammation has also been implicated in the development of depressive disorders. An early lead in the development of the inflammatory theory of depression was the noted incidence of depression-like symptoms in patients receiving IFN immunotherapy. Some studies suggest that as many as 30% of patients developed MDD over the course of therapy, and it was also noted that these symptoms usually decreased once the treatment was completed. As IFN-α demonstrates poor brain penetration, it is unlikely to directly act on the CNS, and instead, the associated release of other pro-inflammatory cytokines and the upregulation of IDO-1 are to blame [48,49]. Increased plasma KYN titres, due to IDO-1 over-expression, are a likely culprit due to KYN’s high BBB permeation and the correlation of CNS and plasma KYN titres, as well as the incidence of depression [8,50]. This suggests that peripheral inflammation may be a significant source of CNS KP metabolites and that IDO inhibition, as a therapy for depression, requires both CNS and non-CNS activities for effectiveness. Further work has demonstrated that IDO upregulation in response to an LPS challenge as well as chronic immune activation can also lead to depression-like behavior in rodent models, indicating that the mechanism of inflammation is not critical to the relationship between inflammation and depression [11,51].

It is important to note that the inflammation theory suffers from the same issues as the monoamine theory. Inflammation should not be considered a universally applicable model of depression; depressive disorder sufferers do not universally present with elevated cytokine levels, nor do elevated cytokine levels universally result in MDD. Additionally, while a positive correlation between cytokine levels and MDD exists, there is an overlap between the groups, indicating that the absolute value of the cytokines is not a strongly useful diagnostic marker. Nonetheless, arguably, there does exist an age-dependent clinical subgroup of depression in which inflammation is a key contributor: geriatric depression.

### Geriatric Depression and Inflammation

In geriatric-aged (i.e., 65 years of age and older) people, depression and medical illnesses are common co-morbidities [52]. While the elderly may not appear more neurochemically prone to depression as a group, it is likely that age-related physical illness and inflammation together are unique contributors to depression in this age demographic. Inflammation, both systemic and regional, increases with age, and inflammatory disorders have greater incidence and prevalence in elderly cohorts [53]. Thus, a case can be made for “geriatric depression” as a phenomenologically distinctive depression subgroup.

As well as the potential for peripheral KP dysregulation driving depression, additional evidence suggests that the aging brain has increased BBB permeability with peripheral immunopeptides/cytokines being more likely to enter the CNS and upregulate IDO-1 in situ, thereby increasing the potential for systemic inflammation and leading to CNS KP dysregulation. A related hypothesis focuses on age-dependent microglial dysregulation, whereby CNS immune changes in the elderly brain lead to altered immune vigilance with microglia responding excessively to cytokine challenges, leading to IDO-1 upregulation and induced KYN formation; this microglial dysregulation is associated with increased depression and suicidality [54,55]. Further support is provided by mouse models with microglial activation, demonstrating increased depressive responses to inflammatory challenges, which can be attenuated by IDO-1 inhibition [56,57]. As such, it is likely that geriatric depression, as a unique age-defined clinical subgroup within the overall spectrum of depressive disorders, may be more likely to benefit from an inflammation-focused intervention.

## 6. IDO and Dementia

Dementia, categorized as a major neurocognitive disorder in the *Diagnostic and Statistical Manual of Mental Disorders* (DSM-5), is a symptom-defined condition characterized by a chronic, progressive deterioration in cognitive function of sufficient magnitude to interfere with the activities of normal daily living [58]. The types of dementia are varied, ranging from the uncommon, such as Creutzfeldt–Jakob disease (CJD), to the more common, such as Alzheimer’s Disease, which accounts for approximately 75% of all dementia cases and currently afflicts more than 50 million people worldwide.

AD manifests clinically as a progressive decline in multiple information-processing domains, including memory, cognition, comprehension, judgment and executive function [59]. This complex symptom cluster arises from AD’s pathology, characterized by cytotoxic misfolded protein oligomerization of β-amyloid (Aβ) and tau, concurrently associated with neurotoxic immuno-inflammation, culminating in concomitant, interconnected yet parallel proteopathic and immunopathic pathogeneses [60]. Initial Aβ oligomerization—beyond being neurotoxic, synaptotoxic and mitochondriotoxic—recruits an extended innate immune response during which C1q (the first subcomponent of the C1 complex of the classical pathway of complement activation) protein co-localizes with Aβ, eliciting a complement cascade and production of inflammatory cytokines and neuroinflammation-associated peptides: IL-1R1, IL-3, IL-4, IL-6, IL-10, IL-12, IL-13, IFN-γ, ICAM-1 (intercellular adhesion molecule 1), MIP-1α (macrophage inflammatory protein-1 alpha), MIP-1β, SDF-1 (stromal cell-derived factor 1) and RANTES (regulated upon activation, normal T-cell expressed and presumably secreted). This further promotes localized microglia-mediated dysregulation of innate immunity as well as toll-like receptor (TLR4) stimulation, inducing NF-κB (nuclear factor kappa B) with the release of TNF-α and IL-1β pro-inflammatory immunomediators. Subsequent inactivation of phosphatases promotes hyperphosphorylation and templated aggregation/propagation of tau protein into synaptotoxic/neurotoxic species, co-operatively augmenting ongoing proteopathic-immunopathic neurotoxicities. Converse activation of the mTOR (mammalian target of rapamycin), GSK3β (glycogen synthase kinase-3 beta) and Cdk5 (cyclin-dependent kinase 5) kinases stimulates multiple neurochemical pathways: increased mTOR signaling enhances Aβ accumulation by decreasing autophagy induction, while amplified GSK3β and Cdk5 crosstalk augments both Aβ/tau- and IL-1β/TNF-α-associated neurotoxicities.

These vicious cycles of immunotoxicity and neurotoxicity render AD an extraordinarily difficult disease to treat; currently, there are no means of curing the disease or halting disease progression. Many years of attempting to devise therapies that target protein misfolding and oligomerization have failed to produce an efficacious agent. Accordingly, immunotoxicity is emerging as a target of interest, and IDO may be a druggable target for the disease modification of AD for a multiplicity of reasons.

In AD, IDO regulates the release of neurotoxic pro-inflammatory cytokines (IL-1β, IL-6, TNF-α) in the brain and influences the proteins (mTOR, GCN2 (general control nonderepressible 2; a serine/threonine-protein kinase that senses amino acid deficiency through binding to uncharged transfer RNA) and eIF2-α (eukaryotic translation initiation factor 2) implicated in AD’s pathogenesis [61]. Furthermore, mTOR plays a role in AD, with its signaling network controlling protein translation, autophagy and synaptic plasticity. Likewise, in AD, GCN2 is a principal kinase shown to phosphorylate eIF2-α protein, blocking protein synthesis and arresting cell growth; conversely, eIF2-α phosphorylation induces the mRNA upregulation of BACE1 (a key enzyme involved in amyloid precursor protein (APP) processing to form Aβ) and ATF4, a repressor of long-lasting long-term potentiation (LTP) that influences synaptotoxicity in AD [62]. Levels of phosphorylated eIF2-α are elevated in the hippocampi of AD patients. The genetic deletion of GCN2 in a transgenic APPswe/PSEN1∆E9 AD mouse prevents impairment of synaptic plasticity and spatial memory, along with decreased phosphorylation of eIF2-α. Similarly relevant to AD, IDO induction modulates the downregulation of brain-derived neurotrophic factor (BDNF) in the prefrontal cortex and hippocampi of mice.

Initial research into the effects of IDO-1 inhibition in the progression of Alzheimer’s has demonstrated some efficacy in the mouse models [63], with AD-type mice demonstrating a recovery of memory in maze tests to the equivalent performance of wild-type mice, as well as a reduction in Aβ plaque burden [64].

### 6.1. Depression-Dementia Spectrum Disorder

The relationship between depression and dementia, specifically AD, is complex across the passage of time and across the network of neurochemical pathways [65,66,67,68]. Depression is a risk factor for AD, and AD is a risk factor for depression. Depression, especially in the elderly, often precedes AD. The two disorders also share a variety of other symptoms, including memory impairment and cognitive decline. It is interesting to speculate that depression (geriatric depression) and dementia (AD) co-exist on a clinical spectrum with significant overlap and mutually triggering attributes. Moreover, though conjectural at the present time, it is likewise interesting to hypothesize that the administration of an antidepressant with an appropriate mechanism of action may delay or prevent progression along the spectrum of dementia. A shared aetiopathogenic mechanism involving elements of the neuroimmune-neuroinflammation axis further suggests that IDO inhibitors may be compounds worthy of therapeutic evaluation for such an indication.

### 6.2. Future of IDO-1 Inhibitors for Treating Dementia and Depression

There is an overwhelming need for new effective treatments for chronic neurologic disorders such as depression and dementia. In the realm of AD and related dementias, the conventional approaches based on the amyloid hypothesis and the inhibition of protein misfolding and oligomerization have failed to yield curative therapeutics [69]. In response to the many failed human clinical trials, new therapeutic avenues are emerging, targeting diverse AD-implicated disease pathologies, including immunopathy, gliopathy, synaptotoxicity, membranopathy, mitochondriopathy, metal dyshomeostasis and reactive oxygen species. Of these many proposed pathological mechanisms, immunopathy, particularly of the innate neuroimmune system, is emerging as a frontrunner.

Likewise, there is a need for effective new agents for the treatment of depression. Despite their disputed efficacy, selective serotonin reuptake inhibitors are widely used across the full spectrum of depressive disorders, regrettably often producing side effects in elderly age groups [70]. Given the clinical uniqueness of geriatric depression combined with the increased prevalence of inflammation in this age group, neuroimmune-mediated neuroinflammation also arises as a viable druggable target for geriatric depression.

Although there are multiple microglial and cytokine-associated targets for the amelioration of neuroinflammation, IDO is a druggable target that, as discussed above, is directly implicated as a key regulator of neuroinflammation. Many years of IDO research in immuno-oncology provide advanced starting points for this research and provide ample justification for pursuing the therapeutic utility of IDO-1 inhibition for pathological neuroinflammation. Moreover, it provides the innovative possibility of assessing IDO-1 inhibitors for depression and their capacity to delay or prevent the subsequent onset of AD.

The future path of IDO-1 inhibitor development within the context of neuroinflammation depends on the design, synthesis and evaluation of safe, brain-penetrant, drug-like small molecules capable of demonstrating a disease-modifying mechanism of action and appropriate target engagement that leads to efficacy in a representative animal model. The data presented in this review suggests that this is possible.

## 7. Other Neurological Disorders

Beyond the time-honored but seemingly failing notion of proteopathic oligomerization, the CNS is also susceptible to damage from other processes, including immunopathy and oxidative stress; accordingly, immunotoxicity and oxidative stress are proposed mechanisms in the pathogenesis of multiple CNS disease states [71]. The role of the KP in the generation of immunotoxins and pro-oxidant species, such as 3-HK and QUIN, increases the putative therapeutic utility of IDO-1 inhibition beyond depression and dementia.

### 7.1. Parkinson’s Disease

As the second most common age-related neurodegenerative disease after AD, Parkinson’s Disease (PD) is characterized by the formation of misfolded α-synuclein protein-rich aggregates, termed Lewy Bodies, and dopaminergic neuronal death. PD typically manifests symptomatically with resting tremors, slowed bradykinetic movements, muscle stiffness and impaired balance. While symptoms can be alleviated by treatment with the dopamine precursor levodopa, there is no cure, and the mechanism of this disease remains incompletely elucidated. Many current theories regarding the pathogenesis of PD mirror those of AD, including elements of protein misfolding/oligomerization, pro-inflammatory immunopathy, mitochondriopathy and oxidative cytotoxicity. As a consensus, immune-mediated neuroinflammation and initial protein misfolding lead to the deposition of Lewy Body proteins, culminating in neurotoxicity, which spurs the further production of toxic reactive oxygen species through a combination of ongoing neuroinflammation and mitochondrial dysfunction [72,73]. Accordingly, immune modulation via IDO inhibition emerges as a justifiable therapeutic approach. IDO-1 inhibition with 1-MT has been examined in mouse models of PD and has shown a generally neuroprotective effect with the restoration of function [74]; mechanistically, multiple routes have been demonstrated for this improvement, including reduced neuroinflammatory bio-markers (TNF-α, IFN-γ and IL-6), mitochondrial dysfunction and neuronal apoptosis (caspase-3) combined with the restoration of neurotransmitter levels (dopamine and homovanillic acid) in the striatum and increased striatal brain-derived neurotrophic factor (BDNF) levels [75]. Overall findings suggest that 1-MT could be a potential candidate for further clinical studies to explore its possibility as an alternative in the pharmacotherapy of PD.

### 7.2. Multiple Sclerosis (MS)

MS is a prototypic chronic neuroimmune disorder [76,77,78,79]. MS is a neurodegenerative condition characterized by the formation of dysmyelinated sclerotic plaques, primarily in brain white matter, which exhibits significant neuroinflammation and microglial macrophageal activity, leading to axonal damage. A typical progression begins with intermittent acute symptoms and may gradually progress over a period of 10 to 15 years. Symptoms typically involve decreased motor and/or sensory function, fatigue and diverse visual symptoms and signs, including optic neuritis, diplopia and internuclear opththalmaplegia. Chronic CNS inflammation is present at all stages of progression, and successful therapeutic approaches have used anti-inflammatory therapeutics to reduce symptoms, which spurred interest in IDO-1 inhibition in the treatment of MS. Paradoxically, however, experimental autoimmune encephalomyelitis (EAE) mouse models that were treated using 1-MT as an IDO-1 inhibitor demonstrated a negative impact on the disease’s progression and reduced recovery. In addressing this unanticipated result, Guillemin and co-workers noted that in the initial phases of the disease, KP activation via IDO-1 is actually beneficial due to facilitating an immune tolerance; however, long-term IDO-1-mediated KP activation may lead to a chronic state characterized by neurodegeneration and that modulating the KP in EAE-induced mice could nonetheless possibly decrease EAE disease severity in the long-term [74]. This remains an area of ongoing study.

### 7.3. Schizophrenia

Schizophrenia is a severe psychiatric disorder prevalent in ~1% of the population. Classic symptoms are hallucination and compulsion; however, schizophrenia also induces negative symptoms, such as reduced emotional range, anhedonia and avolition, as well as cognitive deficits, such as reduced executive function, memory and attention. The discovery of dopamine-suppressing anti-psychotic medications in the 1950s allowed many schizophrenics to function in society. However, despite a half-century of advances in the understanding of the dopaminergic component of schizophrenia, the failure of the dopamine hypothesis to produce more effective drugs or to treat the negative and cognitive symptoms indicates that dopaminergic mechanisms provide only a partial explanation of schizophrenia’s complex underlying aetiopathogenesis.

High levels of pro-inflammatory cytokines have been detected in blood and cerebrospinal fluid samples of people with schizophrenia [80]. Animal models of schizophrenia suggest that an immune disturbance during early life may be a triggering event. Genetic studies have revealed a strong signal for schizophrenia on chromosome 6p22.1, in a region related to the human leucocyte antigen (HLA) system and other immune functions [81]. Zhang et al. demonstrated that an abnormal expression of IDO levels correlated with negative symptoms and pro-inflammatory cytokine levels in patients with first-episode schizophrenia, suggesting the important role of IDO in the pathological mechanism of schizophrenia [80]. Similarly, De Picker et al. showed that the acute schizophrenic psychotic state is marked by state-specific increases in neuroimmune markers and decreases in peripheral IDO pathway markers [82]. Therefore, neuroinflammation, in general, and IDO specifically, may be of research interest in the quest for more effective schizophrenia therapeutics [83]. As with other possible neurologic applications of IDO-1 inhibition, this remains an area of ongoing investigation. Encouragingly, da Silva Araứjo et al. described the reversal of schizophrenia-like symptoms and immune alterations in mice with 1-methyltryptophan by a multifactorial mechanism, including a reduction in hippocampal IL-6 levels and alterations in myeloperoxidase activity and glutathione in the prefrontal cortex and striatum [84].

## 8. Conclusions

IDO-1 inhibition opens a new avenue of approach to treating inflammation-driven neurological conditions and disorders, as well as affording opportunities to synergize with emerging oncological interests in IDO-1-based immunotherapy.

As discussed in this review, IDO-1 inhibition represents a druggable target for the treatment of depression, particularly geriatric depression, and dementia, particularly AD. The inflammatory model of depression provides additional targets for the treatment of depressive disorders, especially in the case of SSRI-resistant depression or geriatric depression. IDO-1 inhibition also has therapeutic potential in the treatment of Alzheimer’s disease by mitigating the impact of amyloid-driven chronic CNS inflammation and the resultant generation of neurotoxic KP metabolites through the upregulated expression of IDO-1. Additionally, KP dysregulation may be a useful approach for the investigation of other forms of inflammation-associated neurological disorders.

The commonality of a disease mechanism (neuroimmunologically mediated neuroinflammation) and a related druggable target (IDO-1) across the two disorders of depression and dementia also raises the interesting possibility of depression (geriatric) and dementia (AD) being a single spectrum disorder. Depression is a risk factor for AD, and many people with AD become depressed; the two disorders co-exist and co-trigger each other. Should such a spectrum disorder exist, it would open the door to other shared therapeutic possibilities beyond IDO-1.

## Figures and Tables

**Figure 1 brainsci-13-00852-f001:**
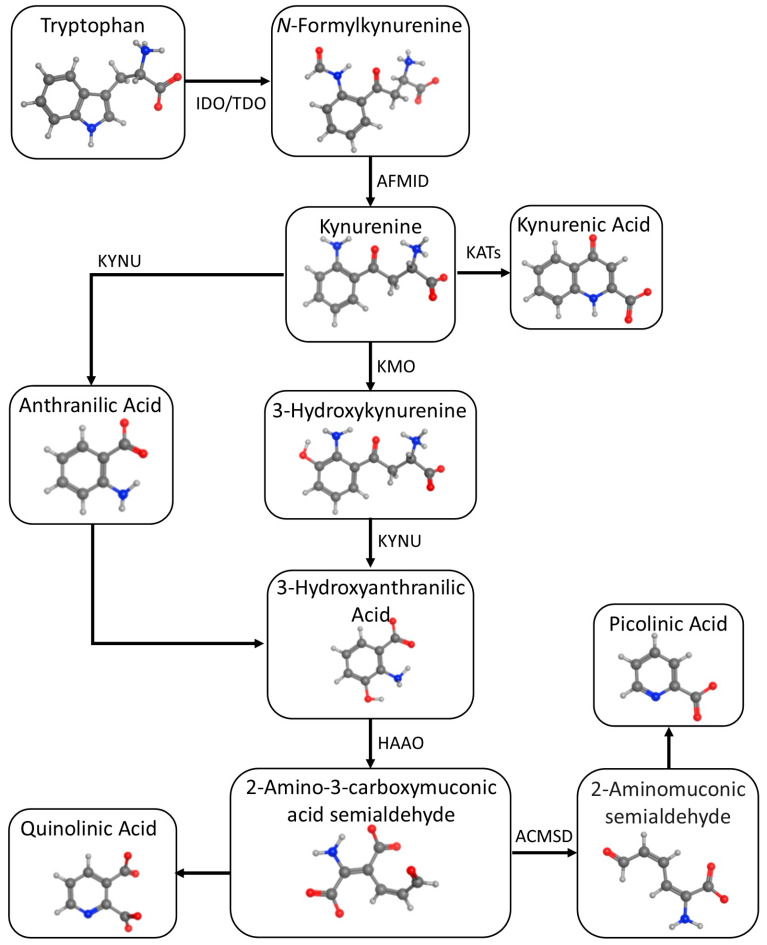
Kynurenic metabolic pathway of tryptophan.

**Figure 2 brainsci-13-00852-f002:**
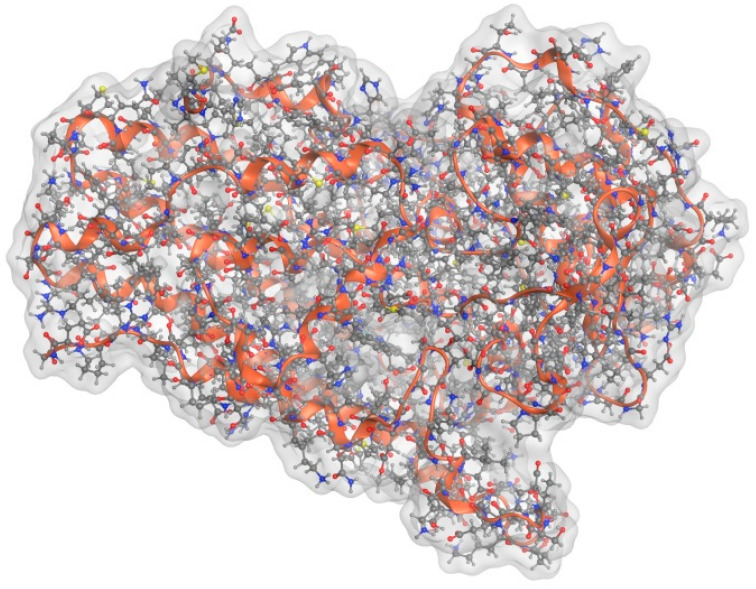
Molecular modeling optimized structure of IDO-1.

**Figure 3 brainsci-13-00852-f003:**
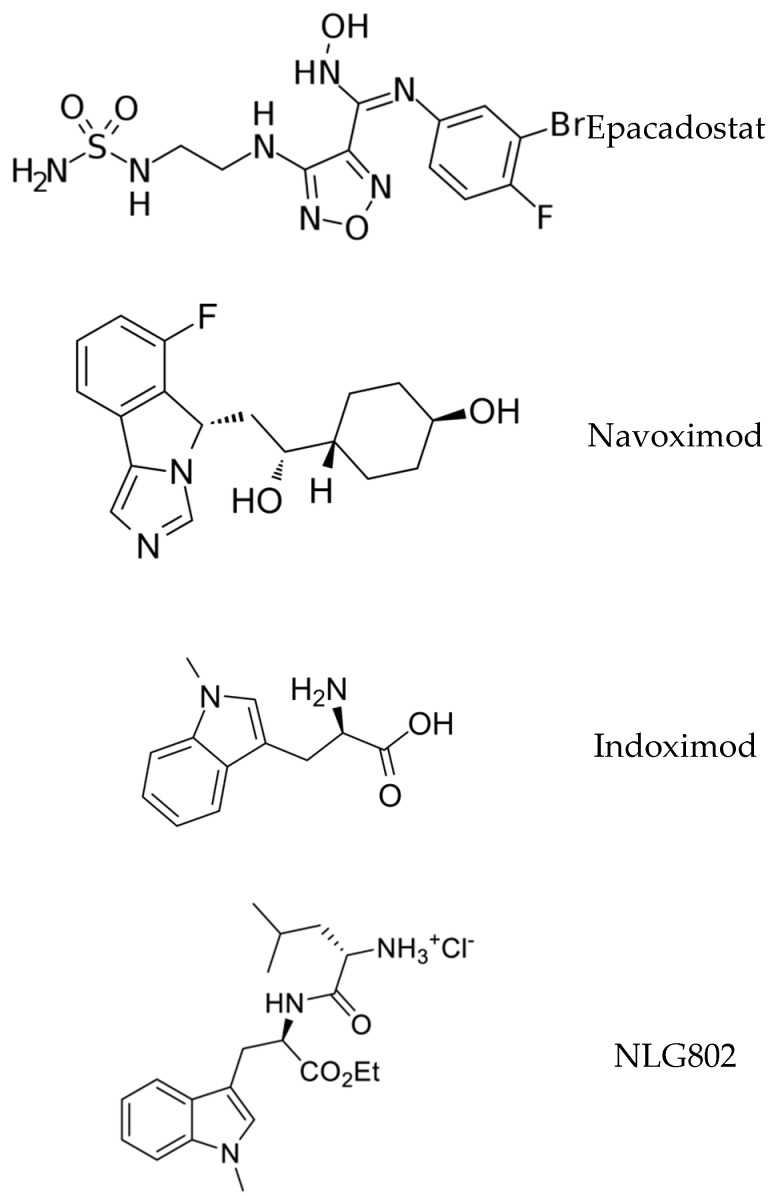
Structures of known IDO inhibitors developed for oncology indications.

**Figure 4 brainsci-13-00852-f004:**
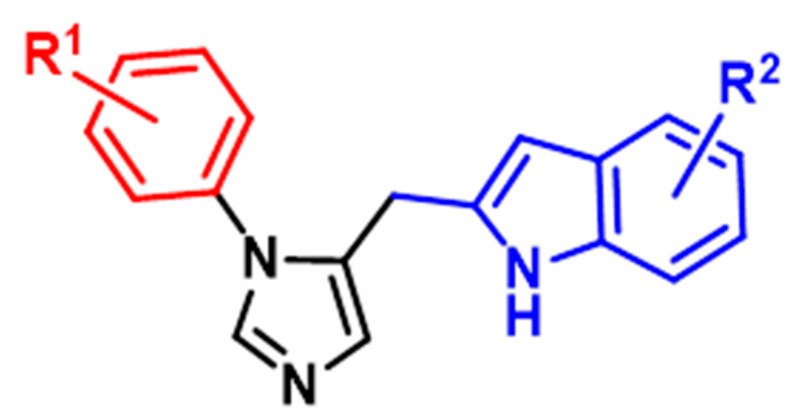
General structure of proposed brain-penetrant IDO inhibitors.

## Data Availability

Not applicable.

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
