# Peer review of "Indoleamine 2,3-Dioxygenase as a Therapeutic Target for Alzheimer’s Disease and Geriatric Depression"

_brainsci, 2023, doi:10.3390/brainsci13060852_

Round 1
Reviewer 1 Report
It is a well written and interesting manuscript. The use of IDO as a drug target may be a promising approach for adjuvant therapy of various types of dementia.
Author Response
Thank you to this reviewer for his comments. No corrections or revisions are suggested.
Reviewer 2 Report
This review focuses on the down-regulation of Indoleamine 2,3-Dioxygenase (IDO-1) as a target against inflammation induced under perturbed neurological conditions, particularly depression, and dement or even cancer. It is a significant contribution to an interdisciplinary scientific community involved in this specific field. It is well-written and the various topics are clearly and consistently presented.
IDO-1 is an inflammatory immune signaling enzyme whose expression is upregulated by proinflammatory cytokines (IFN-γ) and (TNF-α). However,IDO’s potential inhibitors will inhibit specifically the formation of kynurenic compounds without affecting other inflammatory processes leading to cytokine production. This might severely limit the therapeutic benefits.
Specific comments
- Indoleamine-2,3-dioxygenase(IDO-1) is a catalytic enzyme implicated in Tryptophan metabolism, associated with neuroinflammatory effects. It is part of a family of heme-containing enzymes like IDO-1, IDO-2, and TDO, which act to convert Trp to KYN (kynurenic). The review focuses extensively on the role of IDO-1. IDO-2 is less efficient in the catabolizing Trp compared to IDO-1 but can we rule out its contribution to neuroinflammatory effects. Also, the kynurenic pathway produces a large variety of biologically important kynurenic compounds. How different are the kynurenic compounds generated by IDO 1, IDO 2 or TDO?
- Multiple brain-penetrant compounds that interacted favorably with IDO have been designed and synthesized. One analog sounds promising.(IC50 =0.16 μM, EC50 =0.3 μM).More detailed information regarding its structure, its toxicity as well as preliminary assays in AD and geriatric depression would attract the interest of the reader, improve the impact of the review and justify the potential therapeutic benefits.
Author Response
1. This review focuses on the down-regulation of Indoleamine 2,3-Dioxygenase (IDO-1) as a target against inflammation induced under perturbed neurological conditions, particularly depression, and dement or even cancer. It is a significant contribution to an interdisciplinary scientific community involved in this specific field. It is well-written and the various topics are clearly and consistently presented.
Thank you
2. Indoleamine-2,3-dioxygenase(IDO-1) is a catalytic enzyme implicated in Tryptophan metabolism, associated with neuroinflammatory effects. It is part of a family of heme-containing enzymes like IDO-1, IDO-2, and TDO, which act to convert Trp to KYN (kynurenic). The review focuses extensively on the role of IDO-1. IDO-2 is less efficient in the catabolizing Trp compared to IDO-1 but can we rule out its contribution to neuroinflammatory effects. Also, the kynurenic pathway produces a large variety of biologically important kynurenic compounds. How different are the kynurenic compounds generated by IDO 1, IDO 2 or TDO?
In response to this valid criticism, I have added the following paragraph in the Introduction:
Thus, three heme-containing enzymes, IDO-1, IDO-2 and TDO, catalyze the initial rate-limiting step of Trp metabolism leading to the cascade of kynurenine metabolites. Of these three enzymes, IDO-1 is the one for which there are the most data supporting its upregulation during pathological pro-inflammatory conditions. Preclinical and clinical studies support that IDO-1 overexpression is associated with poor prognosis across a spectrum of pathologies, best-demonstrated in cancer [10]. IDO-1 is also widely expressed throughout the brain, particularly in the hippocampal and limbic structures centrally involved in AD [11]. IDO-2 on the other hand is expressed in liver, kidney, and male and female reproductive systems; moreover, as a catalyst, IDO-2 is less efficient and effective than IDO-1 [12]. TDO is also heavily expressed in liver, but has been identified in brain, even co-localized with quinolinic acid, neurofibrillary tau-tangles and amyloid deposits in the hippocampi in both transgenic AD mouse models and human subjects [13]. Because of its well-described upregulation during pathology, brain localization and involvement in AD, IDO-1 has been identified as a possible druggable target for neurodegenerative disorders. A full appreciation of TDO’s participation in neuropathology is still unfolding.
3. Multiple brain-penetrant compounds that interacted favorably with IDO have been designed and synthesized. One analog sounds promising.(IC50 =0.16 μM, EC50 =0.3 μM).More detailed information regarding its structure, its toxicity as well as preliminary assays in AD and geriatric depression would attract the interest of the reader, improve the impact of the review and justify the potential therapeutic benefits.
The description of the structure was improved and Figure 4 has been inserted to provide details about the structure.
Reviewer 3 Report
This is an interesting review on Indoleamine 2,3-Dioxygenase as a Therapeutic Target for Alzheimer’s Disease and Geriatric Depression. The paper is well-written and I agree that it may contribute well to the literature. I only have a few comments to improve the manuscript further:
1. First, I think there should be more elaboration on preclinical and clinical studies of IDO inhibitors which are lacking in the review.
2. Second, it will be useful for the authors to elaborate more on challenge in developing IDO inhibitors as therapeutics.
3. I also hope there will be more elaboration on future directions for IDO research in Alzheimer's disease and Geriatric Depression.
Author Response
First, I think there should be more elaboration on preclinical and clinical studies of IDO inhibitors which are lacking in the review.
The following paragraph has been added:
Preclinical oncology studies evaluating a structurally diverse range of IDO inhibitors showed promising results; for instance, in murine models of melanoma, the combination of IDO-1 blockade with checkpoint inhibitors significantly decreased xenograft growth and increased local cytotoxic T-cell proliferation [39]. Accordingly, multiple IDO-1 inhibitors were advanced to clinical trials with epacadostat receiving extensive study. Although epacadostat exhibits over 1000-fold selectivity for IDO-1 over IDO-2/TDO, the phase III trial of epacadostat combined with the PD-1 inhibitor pembrolizumab did not show clinical benefit when compared with pembrolizumab monotherapy in patients with advanced malignant melanoma – this negative study elicited a general dampening of excitement regarding the therapeutic utility of IDO-1 inhibitors in cancer [40]. Other agents did exhibit some modest efficacies. For example, clinical results indicated that indoximod when used as a single agent, exerted little antitumor efficacy, but when used in combination with other therapies including checkpoint inhibitors (pembrolizumab, nivolumab) and chemotherapy, indoximod demonstrated enhanced antitumour efficacy [41]. Another IDO-1 inhibitor, navoximod, combined with atezolizumab to treat locally advanced or metastatic solid tumors produce a partial response in 9% of dose-escalation patients with adverse events including fatigue, rash, and chromaturia [42]. Overall in the field of IDO-1 inhibition for the treatment of malignancies, there has been a substantive mismatch between preclinical efficacy and ultimate clinical trial outcomes that has significantly reduced widespread development of the therapeutic class.
Second, it will be useful for the authors to elaborate more on challenge in developing IDO inhibitors as therapeutics.
The following paragraph has been added:
Despite the opportunities afforded by these multiple approaches, the development of IDO-1 inhibitors for the treatment of neurological disorders such as depression or dementia must confront a myriad of efficacy and ADMET (absorption, distribution, metabolism, excretion, toxicity) design challenges. The ideal drug should have a half-life of approximately 24 hr. so that it may be administered once per day which is optimal when treating mood or memory disorders. The compound should have minimal interactions with other drugs, since the age group being treated is often receiving multiple medications. Drugs used to treat chronic neurological disorders are typically taken for months or years and thus should exhibit few if any long-term toxicities. The drug must be bioavailable to receptors within the brain and therefore must be able to traverse the blood-brain barrier. Even if the molecule is able to enter brain, will inhibiting IDO-1 be sufficient or should it also be able to concomitantly block TDO. Following receptor(s) binding, the compound should be truly disease modifying and not merely provide symptomatic improvement. Finally, the available animal models for diseases such as AD are best described as models of protein misfolding rather than representative models of the actual disease; therefore, the ability of such animal models to accurately capture the proposed disease-producing mechanism of action of neuroimmune-triggered neuroinflammation is limited constituting a significant impediment to compound advancement. Accordingly, there are many hurdles when attempting to achieve lead optimization and clinical candidate identification in a process that requires proven mechanism of action and evidence of target engagement leading to therapeutic efficacy for IDO-1 inhibitors.
Third, I also hope there will be more elaboration on future directions for IDO research in Alzheimer's disease and Geriatric Depression.
The following new section has been added:
6.2. Future of IDO-1 Inhibitors for Treating Dementia and Depression
There is an overwhelming need for new effective treatments for chronic neurologic disorders such as depression and dementia. In the realm of AD and related dementias, the conventional approaches based on the amyloid hypothesis and the inhibition of protein misfolding and oligomerization have failed to yield curative therapeutics [71]. In response to many failed human clinical trials, new therapeutic avenues are emerging targeting diverse AD-implicated disease pathologies including immunopathy, gliopathy, synaptotoxicity, membranopathy, mitochondriopathy, metal dyshomeostasis and reactive oxygen species. Of these many proposed pathological mechanisms, immunopathy particularly of the innate neuroimmune system is emerging as a frontrunner.
Likewise, there is a need for effective new agents for the treatment of depression. Despite their disputed efficacy, selective serotonin reuptake inhibitors are widely used across the full spectrum of depressive disorders regrettably often producing side-effects in elderly age groups [72]. Given the clinical uniqueness of geriatric depression combined with the increased prevalence of inflammation in this age group, neuroimmune-mediated neuroinflammation also arises as a viable druggable target for geriatric depression.
Although there are multiple microglial and cytokine-associated targets for the amelioration of neuroinflammation, IDO is a druggable target which, as discussed above, is directly implicated as a key regulator of neuroinflammation. Many years of IDO research in immuno-oncology provide advanced starting points for this research and provide ample justification for pursuing the therapeutic utility of IDO-1 inhibition for pathological neuroinflammation. Moreover, it provides the innovative possibility of assessing IDO-1 inhibitors for depression and to assess their capacity to delay or prevent the subsequent onset of AD.
The future path of IDO-1 inhibitor development within the context of neuroinflammation depends on the design, synthesis and evaluation of safe, brain-penetrant, drug-like small molecules capable of demonstrating a disease-modifying mechanism of action and appropriate target engagement leading to efficacy in a representative animal model. The data presented in this review suggests that this is possible.
Reviewer 4 Report
Indoleamine 2,3-Dioxygenase as a Therapeutic Target for Alzheimer’s Disease and Geriatric Depression is interesting and brings relevant points. however the MS is too superficial, and at sometimes hard to follow.
introduction
", and can produced in most biological tissues [5-7]" I think it misses a word here
"with approximately 60% of all brain KYN generated in the periphery external to brain" You should remove the highlighted part, it is redundant with "periphery"
"KYN and 3-HK are distributed bound to plasma proteins and can traverse the BBB, suggesting that increased plasma titres may increase CNS KP dysregulation and promote the formation of QUIN [8]." this sentence is hard to follow I would suggest to modify it.
"was observed that depressed patients exhibit both decreased KYNA:QUIN plasma ratios and decreased cerebrospinal fluid (CSF) KYNA:QUIN ratios, suggesting that in the case of depression there is a positive correlation between peripheral conditions and CNS conditions [12,13]"
and also an increase in QUIN (neurotoxic) if I understand your statement, I think you should mention that.
"This raises the possibility of IDO as a therapeutic target for neuroinflammation associated brain disorders including AD and related dementias as well as MDD" It is not clear how to come to this statement.
I think the introduction is not really clear and should be improved.
Structure and functions of IDO
"IDO-1 also plays a definite role in immune signaling and, as described in mouse models, acts as a pro-inflammatory B lymphocyte mediator [20, 21]. IDO-1 is synthesized as an immune signaling response in situ by multiple cell types [5, 7]. The presence of pro-inflammatory cytokines such as TFN- α, IFN-γ and IL-6 (interleukin-6) upregulates expression of IDO-1 and decreases the local Trp:KYN ratio. This in turn activates Tregulator immune cells, suppressing T-effector cells and dampening the immune response, which may lead to immune escape as evidenced by the unusually high expression of IDO-1 in immune avoidant tumours [22, 23]."
could you develop this section? it is really interesting and need to be develop. IDO-1 acts as a mediator: how? mediator means there is a receptor ? do you have the mechanism?
Druggability of IDO-1
"Accordingly, IDO-1 is a target of interest in oncology, with several drug candidates having gone to Phase I/II human trials in recent years [30, 31]. In addition to the small molecule drug candidates, IDO-1 peptide vaccines have also been designed [32]." could you please develop that. What happened in the clinical trials? and what was the disease?
IDO and dementia
This section would need improvement, you discuss of the disease which is great but in a second part of this section you discuss "In AD, IDO regulates the release of neurotoxic pro-inflammatory cytokines" please give us more concrete information about it. What is the pathway ? IDO is not the only regulating these pathways, which is the dominant?
"IDO induction modulates down-regulation of brain-derived neurotrophic factor (BDNF) in the prefrontal cortex and hippocampi of mice."
This is not clear does it increase or decrease BDNF?
Initial research into the effects of IDO-1 inhibition in the progression of Alzheimer’s has demonstrated some efficacy in mouse models [55], with AD type mice demonstrating recovery of memory in maze tests to equivalent performance as wild-type mice, as well as a reduction in A plaque burden[56]. please develop, how did they induce the inhibition? what is the strain? Only males where used or only females or both? reduction of amyloid where? did they propose a mechanism?
What are your thoughts on this topic? when would you like to modulate IDO in AD: MCI, early stage, late stage? because the inflammation is not present at all stages. Papers demonstrated that the inhibition of the inflammation during the early stage is detrimental.
section 6.1 is not really informative i would suggest to remove it.
section 7
PD:
" IDO-1 inhibition with 1-MT has been examined in mouse models of PD and has shown a generally neuroprotective effect with the restoration of function and improved dopaminergic activity [64]" please develop, this is the most important part of this section.
7.2 MS, please explain or provide insights on why 1-MT failed.
7.3: do you have more examples? this section is superficial
Overall the review is informative but lacks of meaningful info and remains superficial. I would highly suggest to provide more relevant information and discuss more the findings.
Author Response
- Indoleamine 2,3-Dioxygenase as a Therapeutic Target for Alzheimer’s Disease and Geriatric Depression is interesting and brings relevant points. however the MS is sometimes hard to follow.
I agree, particularly in the Introduction which has been reworded in several places as described below.
2. Introduction
" and can produced in most biological tissues [5-7]" I think it misses a word here
a. Yes, you are correct and a word has been added.
"with approximately 60% of all brain KYN generated in the periphery external to brain" You should remove the highlighted part, it is redundant with "periphery"
b. the redundant words have been removed.
"KYN and 3-HK are distributed bound to plasma proteins and can traverse the BBB, suggesting that increased plasma titres may increase CNS KP dysregulation and promote the formation of QUIN [8]." this sentence is hard to follow I would suggest to modify it.
c. this sentence has been rephrased.
"was observed that depressed patients exhibit both decreased KYNA:QUIN plasma ratios and decreased cerebrospinal fluid (CSF) KYNA:QUIN ratios, suggesting that in the case of depression there is a positive correlation between peripheral conditions and CNS conditions [12,13]"
d. I agree, this was very confusing, and this has been reworded.
and also an increase in QUIN (neurotoxic) if I understand your statement, I think you should mention that.
e. done
"This raises the possibility of IDO as a therapeutic target for neuroinflammation associated brain disorders including AD and related dementias as well as MDD" It is not clear how to come to this statement.
f. this final paragraph in the Introduction has been reworked and reworded to justify this statement.
I think the introduction is not really clear and should be improved.
g. I fully agree. I have reworked the Introduction, particularly the final paragraphs. It now reads more clearly as exemplified by this new section now added to the manuscript:
Thus, three heme-containing enzymes, IDO-1, IDO-2 and TDO, catalyze the initial rate-limiting step of Trp metabolism leading to the cascade of kynurenine metabolites. Of these three enzymes, IDO-1 is the one for which there are the most data supporting its upregulation during pathological pro-inflammatory conditions. Preclinical and clinical studies support that IDO-1 overexpression is associated with poor prognosis across a spectrum of pathologies, best-demonstrated in cancer [10]. IDO-1 is also widely expressed throughout the brain, particularly in the hippocampal and limbic structures centrally involved in AD [11]. IDO-2 on the other hand is expressed in liver, kidney, and male and female reproductive systems; moreover, as a catalyst, IDO-2 is less efficient and effective than IDO-1 [12]. TDO is also heavily expressed in liver, but has been identified in brain, even co-localized with quinolinic acid, neurofibrillary tau-tangles and amyloid deposits in the hippocampi in both transgenic AD mouse models and human subjects [13]. Because of its well-described upregulation during pathology, brain localization and involvement in AD, IDO-1 has been identified as a possible druggable target for neurodegenerative disorders. A full appreciation of TDO’s participation in neuropathology is still unfolding.
Since inflammatory conditions upregulate the activity of IDO-1, Trp is increasingly diverted to the KP over the 5-HT pathway in areas of inflammation [14]. Early hypotheses of an inflammatory model of MDD postulated that in chronic or systemic cases of inflammation this diversion of Trp metabolism would cause a relative 5-HT deficiency in the CNS, explaining the onset of depression symptoms. However, more recent data suggest that CNS Trp and 5-HT concentrations in the brain actually increase in response to inflammatory challenge and thus there are other mechanisms at play to regulate 5-HT [15]. However, ongoing data do support a pathogenic role for IDO-1 in MDD. Severe depression is associated with increased microglial quinolinic acid in subregions of the anterior cingulate gyrus – possible evidence for immune-modulated glutamatergic neurotransmission dysregulation [16,17]. IDO-1 blockade in lipopolysaccharide (LPS)-induced rodent models of neuroinflammation has shown reduction of stress and depressive behaviours [15, 18, 19].
This raises the possibility of IDO-1 as a broadly applicable therapeutic target for neuroinflammation-associated brain disorders including AD and related dementias as well as MDD. Since IDO-1 is also a target of interest in neuro-immuno-oncology, in principle lessons learned from the oncology experience and IDO-1 drugs developed for cancer indications could be leveraged as an advanced starting point in the evaluation of the therapeutic utility of IDO-1 inhibition for CNS disease indications such as AD and elderly onset depression.
3. Structure and functions of IDO
"IDO-1 also plays a definite role in immune signaling and, as described in mouse models, acts as a pro-inflammatory B lymphocyte mediator [20, 21]. IDO-1 is synthesized as an immune signaling response in situ by multiple cell types [5, 7]. The presence of pro-inflammatory cytokines such as TFN- α, IFN-γ and IL-6 (interleukin-6) upregulates expression of IDO-1 and decreases the local Trp:KYN ratio. This in turn activates Tregulator immune cells, suppressing T-effector cells and dampening the immune response, which may lead to immune escape as evidenced by the unusually high expression of IDO-1 in immune avoidant tumours [22, 23]."
could you develop this section? it is really interesting and need to be develop. IDO-1 acts as a mediator: how? mediator means there is a receptor ? do you have the mechanism?
a. This section has now been developed further. The following paragraph has been added:
The central role of IDO-1 in regulating immune responses is orchestrated by both enzymatic and non-enzymatic routes. From an enzymatic perspective, the IDO-1 triggered kynurenic pathway contributes to immunoregulation via five mechanisms [27]: [a] by the kynurenic-mediated inhibition of IL-2 signaling impairing memory CD4 T cell survival; [b] by promoting the differentiation of CD4 T cells into Treg cells; [c] by the direct effect of kynurenines on the aryl hydrocarbon receptor (AhR) stimulating dendritic cell differentiation; [d] by modulating Trp via the induction of the stress response kinase, general control nondepressible 2 (GCN2), and the suppression of mammalian target of rapamycin 1 (mTOR1) pathway, which inhibits Teff cell function and maturation; and [e] by the inactivation of the eukaryotic translation initiation factor 2A (eIF2A) which blocks the conversion of Treg cells into pro‐inflammatory T helper type 17 (Th17) cells. The non-enzymatic immunoregulatory function of IDO1 depends on the presence of two immunoreceptor tyrosine‐based inhibitory motifs, ITIM1 and ITIM2, located in the enzyme’s non-catalytic domain [28]. When tyrosine is phosphorylated IDO-1’s ITIM sequence becomes a docking site for various molecular partners containing Src homology 2 (SH2) domains, which enables Src homology 2 domain phosphatase‐1 (SHP1) and SHP2 to interact with the interleukin‐1 receptor‐associated kinase (IRAK), thereby activating noncanonical, anti‐inflammatory NF‐κB rather than canonical, pro‐inflammatory NF‐κB; in turn, noncanonical NF‐κB, associated with activated inhibitory‐κB kinase α (IKKα), translocates into the nucleus inducing the IDO1 gene and subsequently establishing a positive feedback loop conferring a long‐term immunoregulatory phenotype.
4. Druggability of IDO-1
"Accordingly, IDO-1 is a target of interest in oncology, with several drug candidates having gone to Phase I/II human trials in recent years [30, 31]. In addition to the small molecule drug candidates, IDO-1 peptide vaccines have also been designed [32]." could you please develop that. What happened in the clinical trials? and what was the disease?
a. The following paragraph has been added to address these issues:
Preclinical oncology studies evaluating a structurally diverse range of IDO inhibitors showed promising results; for instance, in murine models of melanoma, the combination of IDO-1 blockade with checkpoint inhibitors significantly decreased xenograft growth and increased local cytotoxic T-cell proliferation [39]. Accordingly, multiple IDO-1 inhibitors were advanced to clinical trials with epacadostat receiving extensive study. Although epacadostat exhibits over 1000-fold selectivity for IDO-1 over IDO-2/TDO, the phase III trial of epacadostat combined with the PD-1 inhibitor pembrolizumab did not show clinical benefit when compared with pembrolizumab monotherapy in patients with advanced malignant melanoma – this negative study elicited a general dampening of excitement regarding the therapeutic utility of IDO-1 inhibitors in cancer [40]. Other agents did exhibit some modest efficacies. For example, clinical results indicated that indoximod when used as a single agent, exerted little antitumor efficacy, but when used in combination with other therapies including checkpoint inhibitors (pembrolizumab, nivolumab) and chemotherapy, indoximod demonstrated enhanced antitumour efficacy [41]. Another IDO-1 inhibitor, navoximod, combined with atezolizumab to treat locally advanced or metastatic solid tumors produce a partial response in 9% of dose-escalation patients with adverse events including fatigue, rash, and chromaturia [42]. Overall in the field of IDO-1 inhibition for the treatment of malignancies, there has been a substantive mismatch between preclinical efficacy and ultimate clinical trial outcomes that has significantly reduced widespread development of the therapeutic class.
5. IDO and dementia
"IDO induction modulates down-regulation of brain-derived neurotrophic factor (BDNF) in the prefrontal cortex and hippocampi of mice."
This is not clear does it increase or decrease BDNF?
a. This point has been clarified
6. section 6.1 is not really informative i would suggest to remove it.
a. I disagree with this suggestion. The notion that dementia and depression may co-exist as a spectrum disorder is a potentially important observation that needs to be at least mentioned. I have left this section in.
7. section 7
PD:
" IDO-1 inhibition with 1-MT has been examined in mouse models of PD and has shown a generally neuroprotective effect with the restoration of function and improved dopaminergic activity [64]" please develop, this is the most important part of this section.
The following has been added:
IDO-1 inhibition with 1-MT has been examined in mouse models of PD and has shown a generally neuroprotective effect with the restoration of function [76]; mechanistically, multiple routes have been demonstrated for this improvement including reduced neuroinflammatory bio-markers (TNF-α, IFN-γ, IL-6), mitochondrial dysfunction and neuronal apoptosis (caspase-3) combined with restoration of neurotransmitter levels (dopamine and homovanillic acid) in the striatum and increased striatal Brain Derived Neurotrophic Factor (BDNF) levels [77]. Overall findings suggest that 1-MT could be a potential candidate for further clinical studies to explore its possibility as an alternative in the pharmacotherapy of PD.
7.2 MS, please explain or provide insights on why 1-MT failed.
The following section has been added:
. Paradoxically however, experimental autoimmune encephalomyelitis (EAE) mouse models treated using 1-MT as an IDO-1 inhibitor demonstrated a negative impact on disease progression and reduced recovery. In addressing this unanticipated result, Guillemin and co-workers that in the initial phases of the disease KP activation via IDO-1 is actually beneficial by facilitating immune tolerance, but long-term IDO-1 mediated KP activation may lead to a chronic state characterized by neurodegeneration and that modulating the KP in EAE-induced mice could nonetheless possibly decrease EAE disease severity in the long-term [76]. This remains an area of ongoing study.
7.3: do you have more examples? this section is superficial
An example has been included and is as follows:
As with other possible neurologic applications of IDO-1 inhibition, this remains an area of ongoing investigation. Encouragingly, da Silva Araứjo et al. have described reversal of schizophrenia-like symptoms and immune alterations in mice with 1-methyltryptophan by a multifactorial mechanism including reduction in hippocampal IL-6 levels and alterations in myeloperoxidase activity and glutathione inn the prefrontal cortex and striatum [86].
This reviewer thought the introduction was confusing - i agree and this has been corrected. This reviewer thought that some of the final sections on Parkinson's and schizophrenia were "superficial". I do not fully agree. The focus of this paper is on dementia and depression. However, I felt that it was necessary to demonstrate the full potential impact of IDO-1 in the CNS and deliberately included a very brief overview of several other diseases. I feel that these other diseases add to the overall value of this paper, but are not the focus of the paper and thus have received less detail.
Round 2
Reviewer 3 Report
The authors have addressed my comments well. I appreciate their efforts
Reviewer 4 Report
Thank you for the modifications